# The Relationship between Social Environmental Factors and Motor Performance in 3- to 12-Year-Old Typically Developing Children: A Systematic Review

**DOI:** 10.3390/ijerph18147516

**Published:** 2021-07-14

**Authors:** Dagmar F. A. A. Derikx, Suzanne Houwen, Vivian Meijers, Marina M. Schoemaker, Esther Hartman

**Affiliations:** 1Centre for Human Movement Sciences, University Medical Centre Groningen, University of Groningen, P.O. Box 196, 9700 AD Groningen, The Netherlands; m.m.schoemaker@umcg.nl (M.M.S.); e.hartman@umcg.nl (E.H.); 2Inclusive and Special Needs Education Unit, Faculty of Behavioural and Social Sciences, University of Groningen, Grote Kruisstraat 2/1, 9712 TS Groningen, The Netherlands; s.houwen@rug.nl; 3Department for Human Movement Sciences, University of Groningen, A. Deusinglaan 1, 9713 AV Groningen, The Netherlands; vivian.meijers@live.nl

**Keywords:** motor skills, social environment, social correlates, social interaction, child development

## Abstract

Motor performance during childhood is important for prosperity in life, and the social environment may contain potentially important and modifiable factors associated with motor performance. Therefore, the aim of this systematic review was to identify social environmental factors associated with motor performance in 3- to 12-year-old typically developing children. Four electronic databases were searched, which resulted in 31 included studies. The methodological quality was determined using the Quality of Prognosis Studies in Systematic Reviews tool. Most studies were conducted in 3–6-year-old children. In the home environment, parental beliefs in the importance of physical activity and parental behaviors matching these beliefs were related to better motor performance of children, although these relationships were often sex-dependent. The school and sports environments were investigated much less, but some preliminary evidence was found that being better liked by peers, attending a classroom with a smaller age range, having more interaction with the teacher and classmates, and having a higher educated teacher was related to better motor performance. Further research is required to further unravel the relationship between the social environment and motor skills, with a specific focus on 6–12-year-old children and environments outside of the home environment.

## 1. Introduction

Adequate levels of motor skill performance during childhood are important, because they enable the development of other developmental domains [1]. Furthermore, motor performance may also be predictive for success later in life, such as having a physically active lifestyle [2,3] and academic achievement [4]. However, despite its importance, children’s levels of motor performance have decreased significantly over the past decades [5,6,7]. This places children at increased risk of developing an inactive lifestyle, as well as suboptimal cognitive and social–emotional functioning [2,3,4,8,9]. Therefore, it is important to find ways to stop this trend of decreasing motor performance.

Motor skill performance can be defined as executing learned sequences of movements that can be used to produce a smooth and efficient action in order to achieve an intended outcome [10,11]. Motor skill performance can be subdivided into gross motor skills, which involve the coordination of large muscle groups, and fine motor skills, which involve small movements that require fine precision [10,11]. During the age period of 3–6 years, children acquire and develop basic motor skills required for activities of daily living and interactions with objects and other people [12]. During the age period of 6–12 years, these skills are further refined, and combinations of skills are made, which allows children to participate in sports activities and activities with friends and family [13,14].

Motor development occurs within and in interaction with the environment [15]. According to Newell’s Model of Constraints, motor performance is the consequence of the interaction between the mental and physical characteristics of a person, the task performed, and the environment in which the task is performed [16]. The environment can be divided into the social environment and the physical environment. The social environment includes all social relationships a child may have, while the physical environment refers to all physical components of the area in which a child grows up [17]. Several previous reviews have aimed to identify potential environmental factors that contribute to childhood motor performance in varying age ranges [15,18,19,20]. In these reviews, the focus was mainly on the physical environment of the child while the social environment of the child was overlooked. However, the social environment may contain potentially important and modifiable factors associated with motor performance. People within the social environment of the child, such as parents, siblings, and peers, may influence the motor skills of a child by creating stimulating situations with many movement opportunities and through interactions with the child [21]. During interactions with other people, children may improve their motor skills by observing and imitating the motor performance of others and by practicing their own motor skills at the same time [22,23]. Therefore, in this review, the social environment is defined as the presence of the actors within the social environment, and their behavior that may contribute to the motor development of the child by creating stimulating situations [24]. Furthermore, not only motor skills develop rapidly during childhood, but the social environment evolves as well [19]. During early childhood, the home environment is the most important and influential environment for a child, but when children grow older, they enter new environments such as school and sports environments where they meet new people such as teachers, trainers, coaches, and peers [19]. As both motor skills and the social environment continue to develop, the relationship between them might be susceptible to change as well. Thus, it is important to investigate the relationship between motor performance and the social environment across a broad age range. Knowledge regarding the relationship between motor performance and the social environment might guide parents, teachers, and health professionals in creating situations in which motor skills can be practiced and improved. Therefore, the aim of this review is to identify which social environmental factors have been associated with motor performance in the scientific literature in 3–12-year-old children. 

## 2. Materials and Methods

This systematic review was registered on PROSPERO (registration number CRD42020159935) and was written in accordance with the reporting standards set by the PRISMA 2009 checklist for systematic reviews [25].

### 2.1. Search Strategy

A systematic literature search was performed in March 2020 using four electronic databases (MEDLINE, Embase, PsycINFO, and ERIC). Search terms were created for the three main elements of the research question: (1) children, (2) social environment, and (3) motor performance (see Appendix A for the complete search strategy). Furthermore, search strings were created to include only human studies and to exclude reviews and studies on disabilities and disorders such as cerebral palsy, autism spectrum disorder, and developmental coordination disorder (DCD). The search was further narrowed by using the filter English language only and by setting the publication dates between January 2000 and March 2020.

### 2.2. Selection Criteria

Studies were only included if the relationship between the performance of at least one type of motor skill and at least one social environmental factor was studied in 3- to 12-year-old typically developing children. To be included, the investigated motor skill and social environmental factor had to fall within our definition of these constructs. In this review, motor performance was defined as the execution of learned sequences of movements that can be used to produce a smooth and efficient action in order to achieve an intended outcome [10,11]. The social environment was defined as the presence of actors within the social environment and their behavior that may contribute to the motor development of the child by creating stimulating situations [24]. Studies were only included when participants had a mean age between 3 and 12 years at the time of measurement of motor performance. The age of the participant could be younger than 3 years at the time of measurement of the social environmental factors. The population of interest in this review was typically developing children. Furthermore, only English full-text articles published in peer-reviewed journals were included. Longitudinal studies were only included when participants had a mean age between 3 and 12 years at the time of measurement of motor performance. Intervention studies were only included if the association between a social environmental factor and motor performance was assessed prior to the intervention.

Studies were excluded if the motor performance or the social environmental factor in the study did not fit within our definition of these constructs. Physical fitness and its components did not fall within the scope of this review, and studies investigating physical fitness were therefore excluded. The terms “motor skills” and “physical fitness” are sometimes used interchangeably in the literature, but these constructs are clearly distinct. Physical fitness can be defined as a set of attributes related to the ability to perform physical activity and includes the components cardiorespiratory endurance, muscular endurance, muscular strength, body composition, flexibility, agility, and speed [26,27]. Social environmental factors that occurred prior, during or due to birth (e.g., prenatal or postpartum depression) were excluded. Studies that focused on atypical development were excluded, while studies that focused both on children with a typical and an atypical development were excluded if the groups were not investigated separately. Furthermore, case studies and reviews were excluded from this review.

### 2.3. Data Extraction

After the database search, duplicates were identified and removed in EndNote X9.3.3 using the Bramer method [28]. Two authors (D.F.A.A.D and V.M.) independently screened the titles and abstracts of the studies to assess whether studies were eligible for inclusion; after which, the remaining studies were screened by their full texts to determine whether they met the selection criteria. The same two authors independently extracted the following information from the included studies: study design, age and sex of the study sample, type of motor skill measured, instrument(s) used to measure motor performance, type of social environmental factor measured, instrument(s) used to measure the social environmental factor, and results on the relationship between motor performance and the environmental factor. Any disagreements regarding the study selection and data extraction were resolved through discussion. If a consensus could not be reached, a third author (E.H.) was consulted.

### 2.4. Quality Assessment

The methodological quality of every included study was assessed independently by two authors (D.F.A.A.D and V.M.) using the Quality of Prognosis Studies in Systematic Reviews (QUIPS) tool [29,30]. The QUIPS tool proposes six important domains to assess for potential bias: study participation, study attrition (if applicable), prognostic factor measurement, outcome measurement, confounding measurement, and analysis. These criteria were scored as a low, moderate, or high risk of bias. Any disagreements between the authors were resolved through discussion. If consensus could not be reached, a third author (E.H.) was consulted. The scores on the separate domains were not added together into a total quality score, as this was advised against by the authors of the QUIPS tool [29].

## 3. Results

During the initial database searches, 21,319 articles were retrieved, of which 14,213 remained after removing duplicates. Screening from the titles and abstracts resulted in the removal of 14,020 articles. Cohen’s kappa agreement was 0.513, and the absolute agreement was 98.6%. The remaining 193 articles were screened from the full text; after which, 29 articles were included in this review. The reference lists of all included studies were checked, and two additional relevant studies were identified. These studies were included, which resulted in a total of 31 included studies (see Figure 1).

### 3.1. Included Studies

The extracted data from the included studies are presented in Table 1. Most included studies (83.9%) had a cross-sectional design, or cross-sectional data within the relevant age range could be extracted out of the longitudinal design, while five studies (16.1%) measured the relationship between the social environment and motor performance with a longitudinal design. Three studies measured social environmental factors across multiple time points [31,32,33], and three studies measured the motor performance at two time points [32,34,35].

The ages of the included children ranged from 3 to 12 years, but the majority of the studies were performed on 3–6-year-old children. Only five (16.1%) out of the 31 studies investigated children older than 6 years [36,37,38,39,40]. The included articles measured the performances of different motor skills (see Table 1). For the sake of parsimony, motor performance was divided into fine, gross, and total motor performances for the remains of this paper. The fine motor performance included visual-motor integration and manual dexterity, while the gross motor performance included balance, locomotor performance, object control performance, fundamental motor performance, and bilateral coordination. When no distinction was made between fine and gross motor performance (e.g., psychomotor performance, and perceptual motor performance), this was classified as total motor performance. The social environmental factors that were investigated could be divided into the home environment, which included factors concerning parents and siblings, and the school environment, which included social factors such as the teacher and peers in the class. The sports environment was completely ignored in the previous literature, and social environmental factors concerning the sports environment are therefore not discussed in this review.

**Table 1 ijerph-18-07516-t001:** Overview of the included studies.

Study	Sample Size (♂/♀)	Mean Age ± SD, range	Type of Motor Skill Performance Instrument Used to Measure	Type of Social Environmental Factor Instrument Used to Measure	Result
Barnett et al. 2013 [41]	76 (34/42)	4.1 years ± 0.68, 3.0–6.0 years	Locomotor performance, object control performance Test of Gross Motor Development-2	Parental interaction in child’s PA, parental moderate- and vigorous-intensity PA, parental confidence in their own skills to support child’s activity Parental questionnaire	Parents’ confidence in their own skills was associated with object control performance (ß = 0.23, *p* = 0.038)
Barnett et al. 2019 [31]	178 (sex unspecified)	5 years ^a^	Locomotor performance, object control performance Test of Gross Motor Development-2	Time spent being physically active with mum, time spent with children of a similar age, time spent with older children, parental behaviors (parental facilitation of PA, maternal PA), maternal beliefs (PA optimism, PA self-efficacy) all measured at 4 months, 9 months, 19 months, 3.5 years Maternal beliefs (PA knowledge and PA views measured at 4 months, 19 months, 3.5 years, floor concerns measured at 4 months) Parental questionnaire	Spending time with older children at 3.5 years (ß = 3.00, *p* < 0.05) and maternal optimism at 4 months (ß = 2.43, *p* < 0.05) were positively associated with locomotor performance. Time spent being physically active with mum at 3.5 years (ß = −3.73, *p* < 0.05) and maternal PA at 9 months (ß = −0.01, *p* < 0.05) were negatively associated with locomotor performance. Spending more time with older children at 4 months (ß = 2.27, *p* < 0.05) and 19 months (ß = 2.97, *p* < 0.05) was positively associated with object control performance. Maternal PA knowledge at 3.5 years (ß = −3.05, *p* < 0.05) was negatively associated with children’s object control performance
Bindman et al. 2014 [42]	135 (63/72)	4.56 years ± 0.55, 3.58–5.81 years	Fine motor performance Early Screening Inventory-Revised	Parental graphophonemic support, Parental print support, Parental demand for precision Observation during a writing task with mother	High levels of graphophonemic support were positively associated with fine motor performance (ß = 0.20, *p* = 0.014)
Cao et al. 2014 [33]	89 (42/47)	5.5 years ^a^	Balance, bilateral coordination Bruininks–Oseretsky Test of Motor Proficiency Visual-motor integration The Beery-Buktenica Developmental Test of Visual-Motor Integration	Maternal anxiety measured at 6 months and 5.5 years General Health Questionnaire-28	No significant relationships
Chaves et al. 2015 [37]	390 (186/204)	8.50 years ± 1.27, 6.0–9.99 years	Gross motor performance Körperkoordinationtest für Kinder	School size (number of students) Obtained from school	School size is negatively related to gross motor performance (ß = −0.39, *p* = 0.005)
Comuk-Balci et al. 2016 [43]	437 (sex unspecified)	208 children between 41–56 months, 229 children between 57–80 months	Fine motor performance Denver II developmental screening test	Number of children at home Parental questionnaire	Number of children at home was negatively correlated with 1 out of the 5 fine motor tasks in 41–56 months (r = −0.152, *p* < 0.05)
Cools et al. 2011 [44]	846 (471/375)	5.1 years± 0.6, 4.0–7.0 years	Fundamental motor performance Motor Proficiency Test for 4–6-Year-Old Children	Parental work status (full-time, part-time), family situation (single- or two-parent families), Parental behaviors: parent’s PA, involvement in the children’s play activities, transport habits of the family, and school involvement, Parental beliefs: parental importance rating on developmental and rearing aspects, PA characteristics, and PA equipment characteristics Parental questionnaire	Paternal PA (r = 0.13, *p* < 0.01), father’s involvement in active play (r = 0.11, *p* < 0.05), and parental beliefs on the importance of the child’s PA (r = 0.12, *p* < 0.01), supporting motor development (r = 0.12, *p* < 0.01) and sport specificity of PA (r = 0.19, *p* < 0.01) were related to boys’ fundamental motor performance. Maternal (r = −0.12, *p* < 0.05) and paternal (r = −0.14, *p* < 0.01) school involvement, father’s involvement in creative play (r = −13, *p* < 0.01) and dance activities (r = −0.12, *p* < 0.05), mother’s involvement in gaming (r = −0.14, *p* < 0.05), and parental beliefs on the importance of winning (r = −0.16, *p* < 0.05) were related to girls’ fundamental motor performance
de Oliveira & Jackson 2017 [45]	47 (27/20)	4.67 years ± 0.93, preschoolers ^a^	Fine motor performance Teacher-rated 3-point Likert scale on nine fine motor items taken from normative developmental charts	Rates of maternal verbal support, mother’s encouragement of the child’s autonomy, maternal emotional support, maternal physical support Observation during a building task with mother	In the Somewhat Difficult task, maternal cognitive (r = −0.33, *p* < 0.05) and emotional (r = −0.31, *p* = 0.05) support were negatively correlated to fine motor performance. In the More Difficult task, maternal autonomy support (r = −30, *p* = 0.05) was negatively related to fine motor performance
Fabes et al. 2003 [46]	98 (50/48)	54.77 months ± 10.50, 35–72 months	Perceptual-motor competence Teacher rated five-item scale that measured children’s locomotor, perceptual, and physical skills	Interacting with same-sex peers or in mixed-sex groups Observation during free play	No significant relationships
Giagazoglou et al. 2011 [47]	412 (208/204)	61 months ± 7.7, 4.0–6.9 years	Manual dexterity, object control performance, balance Movement Assessment Battery for Children—2nd Edition	Birth order position Unspecified	No significant relationships
Herry et al. 2007 [48]	821 (406/415)	59.4 months, 48–60 months	Psychomotor performance Early Development Instrument	Number of children per class, family structure (Two- or single-parent households) Questionnaire for teachers and parents	The number of children per class were significantly associated with motor performance
Hua et al. 2016 [49]	4001 (2067/1934)	3.0–6.0 years ^a^	Manual dexterity, object control performance, balance Movement Assessment Battery for Children—2nd Edition	Family structure (single families, nuclear families, extended families) Parental questionnaire Parental rearing behaviors (encouragement of children’s activities/games, teaching verbs, related activities, developing children’s habits, and others) Family Environment checklist on Motor Development for Urban Pre-school Children Class interaction (including amount of supervision, discipline, interaction between teacher and child and interaction between children) Early Childhood Environment Rating Scale–Revised	Parental rearing behaviors were positively related to total motor performance (ß = 0.119, *p* < 0.001), manual dexterity (ß = 0.034, *p* < 0.01), object control performance (ß = 0.062, *p* < 0.05), and balance (ß = 0.024, *p* < 0.05). Class interaction was positively related to total motor performance (ß = 0.139, *p* < 0.01) and balance (ß = 0.184, *p* < 0.001)
Jensen et al. 2019 [50]	130 (sex unspecified)	36 months ^a^	Gross and fine motor performance Adapted version of Mullen Scales of Early Leaning	Maternal distress (stress and depressive symptoms) Bangla version of the Edinburg Postnatal Depression Scale, Perceived Stress Scale Cognitive stimulation Family Care Indicators	Cognitive stimulation correlated to gross (r = 0.216, *p* < 0.05) and fine motor performance (r = 0.186, *p* < 0.05).
Krombholz 2006 [51]	1194 (638/556)	43–84 months	Gross motor performance Forward balancing, hopping on one foot, and 2 items of the Körperkoordinationtest für Kinder (backward balancing, lateral jump) Manual dexterity Paper-and-pencil test	Birth order position Unspecified	Children with older sibling outperformed only or firstborn children on balancing, lateral jump, and hopping on the right foot (no test results)
Kumar et al. 2016 [36]	321	3.0–9.9 years ^a^	Total motor performance Vineland Adaptive Behavior Scale	Joint or nuclear family types, occupation of mother (housewife, working) Parental questionnaire	No significant relationships
Lejarraga et al. 2002 [52]	Heel-to-toe walking: 1182; Copy cross: 996; Draw a person in six parts: 1455	Heel-to-toe walking: 2.83–5.30 years; Copy cross: 3.12–5.20 years; Draw a person in six parts: 3.48–5.93 years	Gross and fine motor performance Score on the developmental items “Copy cross”, “Draw person six parts”, and “Heel-to-toe walk”	Family size, father living at home, birth order position Interview with parents	Birth order position was significant for the fine motor task “Copy cross” (OR = 1.47, 95% CI [1.08, 2.02], *p* < 0.05) and the gross motor task “Heel-to-toe walk” (OR = 0.68, 95% CI [0.50, 0.92], *p* < 0.05)
Lin & Li 2019 [53]	163 (87/76)	38.73 months ± 4.91, 24–47 months	Fine and gross motor performance China Development Scale for Children	Mothers’ play beliefs Chinese Parent Play Beliefs Scale	No significant relationships
Lin et al. 2020 [54]	163 (87/76)	38.73 months ± 4.91, 36–47 months	Fine and gross motor performance China Developmental Scale for Children	Parental play beliefs Chinese Parent Play Beliefs Scale Single child Demographic questionnaire	Children of fathers, who placed a higher value on early academics than on free play, showed poorer gross motor performance than children of fathers who rated free play as more important (F(1, 146) = 3.63, *p* = 0.05)
Livesey et al. 2011 [39]	192 (80/112)	129 months ± 11.1, 105–147 months	Total motor performance Movement Assessment Battery for Children—2nd Edition	Sociometric preference during play and schoolwork Peer Rating Scale Teacher ratings of peer exclusion Peer Exclusion subscale of the Child Behavior Scale	For boys, sociometric preference during play (r = −0.228, *p* < 0.05) and schoolwork (r = −0.245, *p* < 0.05) was related to total motor performance. Peer exclusion was related to total motor performance for both boys (r = 0.447, *p* < 0.01) and girls (r = 0.348, *p* < 0.01).
Lung et al. 2011 [55]	1412 (sex unspecified)	36 months ^a^	Gross and fine motor performance The Taiwan Birth Cohort Study instrument	Maternal self-perceived health status 36-Item Short Form Health Survey Family support, pressure from childcare, number of children in the family, parental marital status Reported by mother	Marriage was a predictor for gross motor performance (β = 0.66, *p* = 0.026). Pressure from childcare (β = −0.11, *p* = 0.011), number of children at home (β = 0.14, *p* = 0.021), family support (β = 0.14, *p* = 0.042), and maternal mental health status (β = −0.02, *p* = 0.047) were predictors of fine motor performance
Luz et al. 2018 [38]	173 (89/84)	8.57 years ± 0.60, 7.00–9.90 years	Gross motor performance Körperkoordinationtest für Kinder	Maternal PA International Physical Activity Questionnaire	Maternal PA was a predictor of motor coordination of girls (OR = 0.183; 95% CI [0.052, 0.642])
Moller, Forbes-Jones, & Hightower 2008 [35]	770 (411/395)	4.15 years ± 0.50, preschoolers ^a^	Total motor performance development over half a year Teacher-rated Child Observation Record	Number of children in a class, range of chronological age in a class Unspecified Range in developmental age in a class Teacher-rated Child Observation Record	Chronological age range (ß = −0.60, *p* < 0.05) and developmental age range (ß = −0.46, *p* < 0.05) were negative predictors of total motor performance development
Moller, Forbes-Jones, Hightower, et al. 2008 [34]	770 (411/395)	4.15 years ± 0.50, preschoolers ^a^	Total motor performance development over half a year Teacher-rated Child Observation Record	Number of children in a class, classroom sex composition Unspecified	No significant relationships
Peyre et al. 2019 [32]	1144 (611/533)	67.8 months ± 1.8, 60–72 months	Motor performance development between 3 and 5 years. A mean score consisting of gross and fine motor performance and visual–motor integration Ages and Stage 2 Questionnaire for gross and fine motor skills; Copy Design task for visual–motor integration	Single-parent household after birth, main caretaker at 2 years, presence of younger and older siblings at 5 years Parental questionnaire Maternal cognitive stimulation measured at 2, 3 and 5–6 years Home Observation for the Measurement of the Environment	Maternal cognitive stimulation at 5 years was associated with motor performance development (β = 0.05, *p* = 0.021)
Sartori et al. 2017 [40]	82 (sex unspecified)	8.5 years ± 0.7, 8.0–9.0 years	Manual dexterity, object control performance, balance Movement Assessment Battery for Children—2nd Edition	Maltreatment and abuse Children were recruited from foster homes after separation from parents due to parental neglect and domestic violence	Children who were maltreated and abused performed worse on balance (F(1.80) = 9.340, *p* = 0.003)
Simcock et al. 2018 [56]	113 (59/54)	48.65 months ± 0.91, 45–51 months	Fine and gross motor performance Ages and Stages 3 Questionnaire	Maternal composite subjective stress Calculated from the Impact of Event Scale-Revised, peritraumatic distress inventory, and peritraumatic dissociative experiences questionnaire Concurrent anxiety, concurrent depression Depression, anxiety and stress scale Marital status (married or de facto vs. single/separated/divorced) Unspecified	No significant relationships
Taverna et al. 2011 [57]	77 (43/34)	53.31 months ± 9.67, 3.0–5.0 years	Gross and fine motor performance Vineland Adaptive Behavior Scales	Culture-sensitive socialization processes (mother social support, father involvement with the family, child autonomy, family connectedness, family involvement in mealtimes) Ecocultural Family Interview	Family connectedness (r = 0.25, *p* < 0.05) and family involvement in mealtimes (r = 0.23, *p* < 0.05) were positively associated with fine motor performance
True et al. 2017 [58]	229 (118/111)	4.2 years ± 0.7, 3.0–5.0 years	Locomotor performance, object control performance, total gross motor performance The Children’s Activity and Movement in Preschool Study (CHAMPS) Motor Skills Protocol	Teacher education Reported by director from preschool	Teacher education was a predictor of total motor score (ß = 0.22, *p* < 0.01) and locomotor performance (ß = 0.14, *p* < 0.001)
Wolf & McCoy 2019 [59]	2137 (1064/1073)	5.16 years ± 1.34, preschoolers ^a^	Fine motor performance International Development and Early Learning Assessment	Caregivers’ cognitive stimulation Six adapted questions from the Multiple Indicators Cluster SurveyCaregiver school-based involvementSelf-reported by caregiver	Caregiver school involvement was a predictor of fine motor performance (ß = 0.08, *p* < 0.05).
Wu et al. 2012 [60]	19,499 (10,237/9262)	3 years ^a^	Fine motor performance The Taiwan Birth Cohort Study instrument	Home environment (cognitive stimulation and emotional support) Adaptation from the Home Observation for the Measurement of Environment Inventory—Short Form	Home environment was associated with fine motor performance (β = 0.05, *p* < 0.001)
Zeng et al. 2019 [61]	100/128	56.08 months ± 4.09	Balance, locomotor performance, object control performance Bruininks–Oseretsky Test of Motor Proficiency, 2nd edition	Number of children in family, parent work status Parental questionnaire	No significant relationships

Note: Reported sample characteristics are from the moment motor performance was measured. The social environmental factors were measured at the same moment unless otherwise stated in the column ‘type of social environmental factor’. Only significant results are presented in the ‘result’ column. Abbreviations: PA = physical activity. ^a^ Age or range of age was not specified exactly but was given in the inclusion criteria for the study.

### 3.2. Quality Assessment

The methodological quality of every included study was assessed across six domains which were scores as low, moderate, or high risk of bias (Figure 2 and Appendix A). Cohen’s kappa agreement and the absolute agreement at this stage were 0.914 and 97.4%, respectively. The domains study participation and study attrition scored the worst. Only 61.3% of the studies were scored as a low risk of bias on study participation, because not all studies described the age (i.e., mean age and age range) and gender distribution of the sample. Only five studies were scored on study attrition, because they had a longitudinal design, of which only two studies (40%) received a score of a low risk of bias. The other domains scored better, with 80.6%, 96.8%, 74.2%, and 87.1% low risk of bias scores on the prognostic factor measurement, outcome measurement, study confounding, and statistical analysis and reporting, respectively.

### 3.3. Home Environment

#### 3.3.1. Parental Characteristics

One of the most investigated social factors regarding the home environment was the family structure. The family structure includes the relationship status of the parents (i.e., married, cohabiting, separated, or single); the composition of the family (i.e., nuclear or joint families); and the involvement of the father in child-rearing. The relationship between these factors regarding family structure and motor performance in 3–6-year-olds was investigated in eight studies. Seven studies found no significant relationships between family structure (i.e., the relationship status of the parents, the composition of the family, and the involvement of the father in child-rearing) and total motor performance [32,44,48,49] or fine and gross motor performance [49,52,56,57]. Only one study found that children with married and cohabiting parents showed better gross motor performance than parents with any other relationship status [55]. One study investigated the relationship between joint or nuclear family types and total motor performance in older children (3–10 years) and also found no significant relationship [36].

A second parental factor that was investigated was parental and maternal work status, which was categorized into full-time working, part-time working, or not working. Two studies investigated the relationship between parental work status and motor performance in 3–6-year-old children [44,61], while one study investigated the relationship between maternal work status and motor performance in 3–9-year-old children [36]. All these studies found no significant relationship between work status and total [36] or gross motor performance [44,61].

Five studies investigated the relationship between the maternal psychological state, which included overall maternal mental health status [55], maternal stress [50,56], maternal depression [50,56], maternal anxiety [33,50,56], maternal social support [57], and maternal perceived pressure from childcare [55], and motor performance in 3–6-year-old children. Two studies found maternal stress and depression not to be related to fine and gross motor performance [50,56]. Maternal anxiety early in the child’s life (at 6 months) was not related to gross and fine motor performance at 5.5 years [33]. Maternal anxiety when the child was 3–6 years old was also not related to fine and gross motor performance [33,50,56]. Another study investigated the social support received by the mother, which was conceptualized as the number of social contacts, amount of support received from those contacts, and how well the mother was coping with everyday life. This study found maternal social support not to be related to fine or gross motor performance [57]. Another study found maternal mental health status and perceived pressure from childcare to be related to fine motor performance but not gross motor performance. Children whose mother perceived more pressure from childcare and had worse mental health had better fine motor performance [55].

#### 3.3.2. Parental Beliefs

Parental behaviors and the way parents raise their children may be influenced by the beliefs parents have. Parental beliefs can be defined as the viewpoints and perspectives parents have about aspects of child-rearing and development [62]. Five studies investigated these parental beliefs in 3–6-year-old children [31,41,44,53,54]. Two studies investigated the relationship between parental play beliefs and motor performance and found that both paternal and maternal play beliefs were not significantly related to fine motor performance [53,54]. Children of fathers who placed a higher value on early academics than on free play showed poorer gross motor performance than children of fathers who rated free play as more important [54]. Children of mothers who rated both free play and early academics as important showed better gross motor performance than children whose mother rated early academics as more important. However, after controlling for variables such as the child’s sex and maternal education level, the relationship between maternal play beliefs and motor performance was no longer significant [53]. Another study investigated maternal physical activity (PA) beliefs such as knowledge regarding the importance of PA, PA views, optimism in engaging children in PA, confidence in promoting PA, and perceptions of the safety of floor play during multiple time points in the child’s life (4 months, 9 months, 19 months, and 3.5 years). Maternal optimism measured at 4 months was positively related to gross motor performance measured at 5 years, and more knowledge about the importance of PA measured at 3.5 years was associated with a worse gross motor performance at 5 years. None of the other relationships were significant [31]. Another study examined parental beliefs by asking parents how much importance they placed on several developmental aspects and PA characteristics [44]. This study found that, when parents thought it was important for their child to participate in PA, work towards sport-specific goals, and thought PA should support the motor development of the child, it was related to a better gross motor performance of boys, while the parental belief on the importance of winning was negatively related to the gross motor performance of girls [44]. The last study investigated the relationship between parental confidence in their own skills to support their child’s activity and motor performance and found that higher parents’ confidence in their own skills was associated with better object control performance of the child but not with locomotor performance [41].

#### 3.3.3. Parental Behaviors

An important variable to investigate when looking at the home environment is parenting behavior. Parenting behaviors can be defined as parenting practices or approaches to child-rearing [62]. Four studies investigated whether cognitive stimulating activities (e.g., reading, storytelling, singing, counting, playing, and going outside) in the home environment were related to motor performance in 3–6-year-old children. Three studies found that more stimulation at home between the ages of 3–6 years was related to better fine [50,60], gross [50], and total motor performance [32], while another study found no relationship between the number of stimulating activities and fine motor performance [59]. The amount of cognitive stimulation provided earlier in a child’s life (i.e., at 2 and 3 years) was also not related to the total motor performance at 5- to 6-years-old [32]. 

Other parental behaviors that were investigated were parental PA behaviors [31,38,41,44] and rearing behaviors [44,49,57,59]. Three studies investigated the relationship between the amount of PA performed by parents and motor performance in 3–6-year-old children. One study found that the amount of parental moderate- and vigorous-intensity PA was not related to gross motor performance [41]. Another study measured the amount of maternal PA at different time points in the child’s life (4 months, 9 months, 19 months, and 3.5 years) and measured the gross motor performance at 5 years [31]. Only a higher amount of maternal PA at 9 months was associated with poorer gross motor performance at 5 years. The third study found that the amount of paternal PA was associated with a better total motor performance of boys but not girls, while the amount of maternal PA was not related to the gross motor performance of either boys or girls [44]. One study investigated the relationship between the amount of parental PA and motor performance in older children (>6 years). In this study, more maternal PA was associated with a better total motor performance of 7- to 9-year-old girls but not boys [38]. Not only the amount of parental PA but, also, the interaction between parents and the child during PA might be related to the child’s motor performance. One study found that the parental interaction with the child during PA was not significantly associated with gross motor performance [41]. Another study investigated the relationship between the time spent being physically active with the mother while the child was 4 months, 9 months, 19 months, and 3.5 years old and the gross motor performance at 5 years [31]. Children who spent the most time being physically active with mum at 3.5 years had poorer gross motor performance. Only one study investigated whether the parental facilitation of PA throughout the child’s life (at 4 months, 9 months, 19 months, and 3.5 years) was related to the gross motor performance at 5 years but found no significant relationship [31]. One study found that parental rearing behaviors, constructed as one score, including the encouragement of children’s activities/games, teaching verbs, and developing children’s habits, was positively related to the fine, gross, and total motor performance of 3–6-year-old children [49]. Three studies investigated several rearing behaviors separately [44,57,59]. The first study investigated the involvement in school; transport habits; and involvement in activities such as playing, reading, gaming, and TV viewing separately and found no significant relationship with the gross motor performance of 4- to 6-year-old boys [44]. For 4- to 6-year-old girls, more paternal school involvement was related to better total motor performance, while more paternal involvement in creative activities and more maternal involvement in gaming led to worse gross motor performance [44]. Another study reported more caregiver school involvement to be predictive of the better fine motor performance of 3–6-year-old children [59]. Other rearing behaviors that were investigated in relation to fine and gross motor performance were mother social support, child autonomy, family connectedness, and family involvement in mealtimes [57]. Only more family connectedness and family involvement in mealtimes were related to the better fine motor performance of 3–5-year-old children, while none of these behaviors were related to gross motor performance [57]. 

Two studies investigated the relationship between the parental support provided during the execution of a fine motor task and fine motor performance in 3–6-year-old children [42,45]. The first study involved a writing task during which graphophonemic support (i.e., help with isolating sounds within words to match them with corresponding letters), print support (i.e., help with producing letter forms on paper), and demand for precision (i.e., the degree to which parents point out errors) were measured. Children who were given more graphophonemic support showed better fine motor performance, while print support and demand for precision were not related to fine motor performance [42]. The other study consisted of two building tasks with different task difficulties during which the rates of cognitive, emotional, physical, and autonomy support were measured. More maternal cognitive support (i.e., explaining the task) and emotional support (i.e., praising the child) during the easier task, as well as more maternal autonomy support (i.e., encouraging the child to think for her- or himself through the use of facilitative questions) during the more difficult tasks were related to poorer fine motor performance [45]. Finally, one study investigated the relationship between maltreatment and abuse of the child and motor performance and found that 8-year-old children who were previously maltreated performed significantly worse on balance but not on manual dexterity and object control performance [40].

#### 3.3.4. Siblings

Two studies investigated whether the presence of siblings was related to fine and gross motor performance in 3-year-olds [54] and in 5–6-year-olds [32] and found no significant relationships. Another study investigated if spending time with older children in the family and home environment, not specifically limited to siblings, at different time points in life was related to motor performance at 5 years of age. Spending time with older children at 19 months and at 3.5 years resulted in better locomotor scores, while spending time with older children at 4 and 19 months was predictive of object control performance [31]. Four studies investigated the relationship between the number of siblings and motor performance in 3–6-year-old children [43,52,55,61]. The three studies investigating gross motor performance found no relationship with the number of children at home [52,55,61]. The three studies investigating the relationship between the number of siblings and fine motor performance found mixed results. One study found no significant relationship [52], another study found a significant relationship in only one out of the five fine motor tasks measured [43], while a third study found that the number of children at home was a significant predictor of fine motor performance [55]. Another variable that was investigated with regards to siblings of 3–6-year-old children in relation to motor performance was the birth order position of the child. The three studies investigating this variable showed mixed results [47,51,52]. One study found that firstborn children performed better on one out of the two fine motor tasks and on the only gross motor task than later-born children [52], while another study found that children with older siblings outperformed the firstborn children on gross motor performance but found no difference in fine motor performance [51]. The last study found no significant relationship between the birth order position of the child and fine, gross, and total motor performance [47].

### 3.4. School Environment

Besides the home environment, the school environment is an important place for children to have social interactions with, for example, peers. One study investigated the relationship between the sex of peers and motor performance in 3–5-year-old children and found that the proportions of same-sex and mixed-sex play were not related to the teachers’ perceptions of preschoolers’ total motor performance [46]. The relationship between peer preference and peer exclusion, on the one hand, and motor performance on the other hand was investigated in 9–12-year-old children [39]. Peer preference, both during schoolwork and play, was negatively correlated to motor impairment scores for boys but not for girls, meaning that boys who were less liked by their peers showed poorer total motor performance. Teacher ratings of peer exclusion were positively correlated with motor impairment scores for both boys and girls, meaning that children who were more excluded by peers showed poorer total motor performance [39].

Other studies investigated variables related to the class and school environment, such as the number of students in relation to motor performance. One study found that the number of students per class was significantly related to total motor performance of 3–6-year-old children [48]. However, this study did not mention the direction of this relationship, and therefore, it is not clear whether the class size was positively or negatively related to the total motor performance. Contradictory, two studies found that the number of students per class was not significantly related to the total motor performance scores of 3–6-year-old children [34,35]. However, it should be mentioned that these studies included the same sample twice to investigate several class-related variables. Therefore, these aforementioned results did not confirm each other but, rather, repeated each other [34,35]. Another study investigated the school size and found that the higher the number of students in a school, the lower the gross motor performance of 6–10-year-old children [37]. Two studies looked at the classroom composition and found that both the chronological age range and developmental age range were negatively associated with the development of the total motor performance of 3–6-year-old children over half a year, meaning that children in a class with a larger chronological and developmental age range had a poorer development of their total motor performance [35]. The sex composition within a classroom was not related to the development of total motor performance [34].

Another important factor to consider in the educational environment is the teacher. One study found that a higher percentage of classroom teachers with a college degree was related to the better total motor performance and locomotor performance of 3–6-year-old children but not to object control performance [58]. Another study looked at classroom activity interactions, which were conceptualized as one variable composed of the amount of supervision, discipline, interaction between teacher and child, and interaction between children [49]. A higher level of classroom interaction resulted in the increased gross and total motor performances of 3–6-year-old children.

## 4. Discussion

The aim of this review was to identify the social environmental factors associated with motor performance in 3–12-year-old typically developing children. The social environmental factors that were investigated could be subdivided into the home environment, which included factors concerning parents and siblings, and the school environment, which included social factors such as the teacher and peers in the class. The results related to parents were further subdivided into parental characteristics, parental beliefs, and parental behavior. Parental characteristics were not related to motor performance, while parental beliefs regarding PA participation and motor development were related to motor performance, although these relationships were often sex-dependent. Furthermore, mixed results were found for the relationship between parental behaviors and motor performance, and some of the relationships found were sex-dependent. Only a relatively low number of studies investigated the relationship between the presence and behavior of siblings, peers, and the educational setting on the one hand and motor performance on the other hand. These studies resulted in mixed findings for the relationship between the presence and behavior of siblings and motor performance. Furthermore, some preliminary evidence was found that children who were better liked by peers, were in a classroom with a smaller age range, had more interaction with the teacher and classmates and had a teacher with higher education, showed better motor performance.

### 4.1. Home Environment

#### 4.1.1. Parents

One of the most important environments for a child is the home environment, and parents play an important role in shaping this environment [62]. According to the social learning theory, behaviors are shaped through interactions with others and by imitating the behaviors of others [22]. Vygotsky’s developmental theory supplements this by stating that social interactions are fundamental to development and that especially older and more competent people, such as parents, can serve as an example and guide children through new tasks [63]. Most of the results of the studies included in this review concerned parents, and these results could be subdivided into parental characteristics, parental beliefs, and parental behaviors. A previous study, outside the scope of this review, suggested that these three domains form the home learning environment and proposed a framework on how the home learning environment is related to a child’s development [64]. This framework was designed for a child’s overall development but can also be used to explain the relationship between the home environment and motor performance (Figure 3). Parental behaviors, which can be defined as parenting practices or approaches to child-rearing [62], are directly related to motor performance. Parents can create a stimulating home environment through their behaviors by participating in stimulating activities and by providing learning materials [21,65]. Parental characteristics, which can be defined as the background of the parents, and parental beliefs, which can be defined as the viewpoints and perspectives parents have about aspects of child-rearing and development, are only indirectly related to motor performance mediated by parental behaviors [62,64]. In other words, parental characteristics are related to parental behaviors, because these characteristics might affect how much time and effort parents are able to put into their child’s development [65]. Furthermore, it can be assumed that parental beliefs and how much value they place on the different aspects of child-rearing and development influences their behavior [64]. The results concerning parents found in this review will be discussed in light of this framework.

Twelve of the included studies in this review investigated parental characteristics. Most of these studies found no relationship between family structure [32,44,48,49,52,56,57], parental work status [36,44,61], and maternal mental health [33,50,56,57] on the one hand and motor performance on the other hand. These results are in line with studies investigating other developmental domains, which found parental characteristics to be less strongly correlated with a child´s intellectual and social development than parental behaviors [66]. In other words, parental behaviors (i.e., what parents do with their children and which activities are performed together) seem more important in a child’s development than parental characteristics. The same may apply to motor performance, and this would be in accordance with the aforementioned framework, in which parental characteristics are only indirectly related to a child´s motor performance, with parental behaviors as a mediating factor [64]. Furthermore, only three parental characteristics were investigated in the included studies, and it might be that these specific characteristics are not related to motor performance. Other parental characteristics that might influence how much time and effort parents put into creating a stimulating environment with sufficient movement opportunities may be the cognitive and motor abilities of parents and their ethnicity. Therefore, further research is required to draw conclusions about the relationship between parental characteristics and motor performance.

According to the framework described before, parental beliefs also have an indirect relationship with a child’s motor performance, with parental behavior as a mediating factor [64]. Five studies included in this review investigated the relationship between parental beliefs regarding play [53,54], PA, and motor development [31,41,44] on the one hand and motor performance on the other hand. The two studies investigating parental play beliefs found that children whose parents placed a higher value on free play had better gross motor performance, although this relationship was no longer significant for maternal play beliefs after controlling for the child’s sex and maternal education level [53,54]. Parents who endorse free play probably give their children more opportunities to engage in free play, during which children can practice their gross motor skills [53]. The two studies investigating the relationship between motor performance and several beliefs regarding the importance of motor development and PA-related beliefs and motor performance found no relationships when investigating both boys and girls together [31]. However, when the sexes were investigated separately, parental beliefs regarding the importance of PA participation and working towards sport-specific goals and the belief that PA should support the motor development of a child were related to the better gross motor performance of boys, while only the belief in the importance of winning was related to a poorer gross motor performance of girls [44]. The relationships found between the parental beliefs and better gross motor performance of boys confirm the hypothesis that parental beliefs are related to motor skills [64]. Parents who believe in the importance of PA and motor development probably encourage and stimulate children in these areas, which explains the superior motor performance. Furthermore, these results also suggest that parental PA and motor development-related beliefs may be sex-dependent. This is supported by a previous study, in which the interaction with parents was found to be beneficial for the fundamental motor skills of boys, while girls benefited from independence [67]. It is still unknown why the role of interactions with parents may be different for boys and girls, but one explanation might be that parents spend more time in gross motor activities with their sons while participating in more stationary activities with their daughters [68,69]. Only one parental belief—namely, the importance of winning—was related to the worse gross motor performance of girls [44]. When parents believe that it is important that their children are the best in sports, this may put pressure on a child. This is further corroborated by the finding that more parental pressure was related to less motivation and enjoyment of children in performing sports, which eventually leads to less participation in sports [70,71]. Therefore, when parents believe that winning is important, this may create an unstimulating environment to practice motor skills. It is not clear yet why especially girls are susceptible to these negative consequences.

Lastly, several parental behaviors were investigated in relation to motor performance. The majority of the studies found children, whose parents provided cognitive stimulation between the ages 3–6 years, to display better motor performance [32,50,60]. Moreover, overall rearing behaviors, constructed as one total score [49], and school involvement of the caregiver or parents [44,59] were related to better motor performance. However, other studies found mixed results regarding rearing behaviors [44,57] and PA behaviors [31,38,41,44]. One striking similarity was found in the studies that found mixed results, as no relationships were found when boys and girls were investigated together [31,41,57], while some relationships were found when boys and girls were investigated separately [38,44]. These results suggest that parental rearing and PA behaviors, just as some of the parental beliefs, may be sex-dependent. One study investigating several rearing behaviors found none of these behaviors to be related to the gross motor performance of boys, while more paternal involvement in creative activities and more maternal involvement in gaming was found to be related to a worse gross motor performance of girls [44]. These findings again corroborate that boys’ fundamental motor skills may benefit from interactions with parents, while girls’ fundamental motor skills may benefit from independence [67]. Studies investigating the relationship between the amount of parental PA and children’s motor performance by sex, found more paternal PA to be related to better gross motor performances of 4–7-year-old boys, while the amount of maternal PA was not related to the gross motor performance of either boys or girls [44]. Meanwhile, another study found more maternal PA to be related to a better gross motor performance of 7–9-year-old girls [38]. These results indicate that maternal and paternal role modeling are potentially sex-specific, with boys following their father’s example and girls following their mother’s example. A study investigating parent–child PA correlations by sex in 10-year-old children found similar results, with positive associations between fathers’ and boys’ PA and between mothers’ and girls’ PA [72].

#### 4.1.2. Siblings

The relationship between the presence of siblings and motor performance was only investigated in 3–6-year-old children and resulted in mixed outcomes. The presence of siblings was not related to either fine or gross motor performance [32,54], while spending more time with older children, not specifically restricted to siblings, throughout early childhood was related to the better gross motor performance of 5-year-old children [31]. The number of siblings was not related to gross motor performance [52,55,61], while mixed results were found for fine motor performance [43,52,55]. The studies investigating the relationship between birth order position and motor performance found mixed results for both gross and fine motor performances [47,51,52]. Two explanations are often used to describe the relationship between the siblings and motor skills of children. The relationship may be positive, because siblings can act as role models and provide a safe environment in which motor skills can be practiced [15,73]. The relationship could also be negative, because siblings might also act as competitors for parents’ time and care. This decreases the opportunities for interactions between the child and parents, which may negatively impact their motor performance [73,74]. However, a third explanation, that might explain the mixed results, could be that the relationship can be either positive or negative depending on the situation of the child [74]. A sibling might act as a role model, but if the sibling has poor motor skills, then the child will not gain much by imitating these motor skills [22,75]. Furthermore, the presence of more siblings might lead to a dilution of parental resources, but it may also provide learning experiences for parents that give them more experience in raising children [74].

### 4.2. School Environment

School is an influential place for children, as they spend many hours a day at school. Besides parents and siblings, children spent an important amount of time with peers. The sex of the peers the children played with was not related to the motor performance in 3–5-year-old children [46]. This was a surprising result, because the way of playing between same- and mixed-sex at this age is completely different, as shown in a study that investigated how preschoolers played as a function of their own sex and the sex of their play partner. Boys’ same-sex play mainly consisted of physical play, while the girls’ same-sex play mainly consisted of pretend play. However, when children played with children of the other sex or in a group that contained both sexes, they engaged in both physical and pretend play [76]. Due to these differences in play preferences, with which different types of motor skills were practiced, it was expected that the proportions of same- and mixed-sex play would be related to motor performance. It is not clear yet why this was not the case. Other important factors concerning peers in relation to motor performance were peer preference and peer exclusion. The relationships between peer preference and peer exclusion on the one hand and motor performance on the other hand were investigated in 9–12-year-old children. Boys who were more liked by their peers showed better motor performance, while both boys and girls who were more excluded by peers showed poorer motor performance [39]. These results show that the relationship between social interaction with peers and motor skills may be bidirectional [77]. Children may improve their motor performance through interactions with peers by observing and imitating their motor performance and practicing their own motor performance, as described before. However, the level of motor performance may also influence if and how children interact with peers. Previous studies in 5–11-year-old children with DCD found worse motor performances when parents reported more peer problems such as aggressive behavior [78], and these children were observed to interact less with peers on the playground [79].

Regarding the school environment, the variables class and school size, teacher education, class composition, and interactions with the teacher and other children were investigated. Mixed results were found for the relationship between class size and total motor performance in 3–6-year-old children. Two studies, investigating the same study sample, found no relationship [34,35], while another study found a relationship but did not define the direction of the relationship [48]. Previous research suggests that a higher number of students in a class might be related to poorer motor skills, because the available resources such as space to move around and teachers’ time and attention are more diluted [58,80]. However, a review has shown that the effect of class size on other domains such as academic achievement is small at best [80]. Therefore, it might be that the influence of class size on motor performance is small and dependent on other factors such as available space and equipment. Notwithstanding, a higher number of students in school was related to poorer gross motor performance in 6–10-year-old children [37]. A possible explanation for this might be that the school size is coupled to the type of area (e.g., rural or urban). In urban areas, the population density is higher, and school sizes are generally larger [81]. However, the movement opportunities in urban areas might be sparser than in rural areas [82]. Therefore, a bigger school size might be related to poorer gross motor skills, and this relationship might be mediated by the type of area the school is in. 

Regarding the composition of the classroom, both the sex and the age composition were investigated in relation to the change in total motor performance of 3–6-year-old children over half a year. The sex composition was not found to be related to motor performance [34], which fits the finding earlier described that sex of the peers that the 3–5-year-old children played with was not related to motor performance [46]. A wide range of ages in a classroom was related to poorer motor performance, meaning that a wider range in both chronological and developmental ages was associated with poorer motor performance [35]. This finding was surprising, as it was expected that older (i.e., higher chronological age) and more competent (i.e., higher developmental age) children could serve as an example for younger children, thereby creating an optimal environment for motor development [63]. However, this positive effect for the younger children may implicitly result in negative consequences for the older children, as they do not have older and more competent children to model [35]. Another pathway through which the age range in a class might negatively relate to motor skills is through the teacher. It is likely that it costs more time and attention of the teacher to instruct and interact with children in a class with a broader age range, which then cannot be invested in the children individually [35].

Not many variables with regards to the teacher were investigated, but one study found that a higher percentage of classroom teachers with a college degree was related to better total motor performance and locomotor performance of 3–6-year-old children but not to object control performance [58]. A previous review found that classroom teachers with a higher education level provided a better-quality learning environment for 3–5-year-old children than teachers with less education [83]. Although it is unclear why teacher education was not related to object control performance, these results suggest that teachers with a higher level of education are better able to provide optimal preschool environments with sufficient movement opportunities. The last variable that was investigated was classroom interactions, which was conceptualized as one variable composed of the amount of supervision, discipline, interactions between teacher and child, and interactions between children [49]. A higher level of classroom interaction resulted in increases in the balance and total motor performance of 3–6-year-old children. This finding supports the social learning theory that states that behavior is shaped through interactions with others and by imitating the behaviors of others [22,23].

### 4.3. Future Directions

For this review, databases were searched for studies between 2000 and 2020. Out of the 31 studies that were found, 25 studies were published after 2010. This suggests that research into the relationship between motor performance and the social environment is a relatively new area of interest, which explains why there is such a relatively low number of studies performed on such an important and comprehensive topic. Looking at the results, it was striking that only four studies investigated the relationship between motor performance and the social environment in children older than 6 years [36,37,38,39,40]. However, the development of motor skills, albeit less rapidly, still continues between the ages of 6–12 years, and therefore, these years are important as well [13,14]. The social environment evolves as well as children grow up, because they enter new environments where they meet new people, such as teachers, trainers, coaches, and peers [19]. It is therefore important to investigate the relationship between motor performance and the social environment across the entire age range during which motor development occurs and not just during the younger years. Another striking result is the imbalance in studies investigating the different social environments. Most studies investigated the home environment and mainly focused on the parents, while the school environment was mostly ignored, and the sports environment was completely ignored. This imbalance in studies might be explained by the imbalance in age, as mostly children younger than 6 years were investigated, for whom the home environment is the most important environment [19]. However, children spend a significant portion of their days in school and sports environments, and therefore, these are important to take into account as well. Furthermore, the interactions between the different social environments, such as the home and the educational environments, have not been investigated at all. The latter might be important, because the different social environments might influence each other and the child´s development in their own way [84]. Fewer interactions or a complete lack of interactions with one actor in the social environment might be compensated for by interactions with other actors in the social environment, or it might negate the influence of interactions with other actors. For example, an only child might compensate the lack of interaction with siblings by seeking more interactions with parents or peers. In conclusion, the relationship between the social environment and children’s motor performance is quite a new area of research, and further studies are required to unravel these relationships. Future research should especially focus on children in the age range of 6–12 years and should not only focus on the home environment but, also, on the environments outside of the home, such as school and sports clubs. Furthermore, it might be interesting to investigate the relationship between the social environment as a whole and motor performance.

## 5. Conclusions

The results yielded in this systematic review could be divided into the home environment and the school environment. Most included studies on the home environment focused on parental factors and found that children whose parents believed in the importance of physical activity and behaved accordingly were found to have better motor performance, although these relationships were often sex-dependent. Studies investigating the relationship between the presence of siblings and motor performance resulted in mixed outcomes. The school environment was investigated much less, but some preliminary evidence was found that children who were better liked by peers—attending a classroom with a smaller age range—had more interactions with the teacher and classmates, and had higher-educated teachers showed a better motor performance. In conclusion, the results of this review showed preliminary evidence for relationships between the presence and behaviors of people within the home and school environment and the motor performance of a child. This suggests that a stimulating environment, in terms of interactions with others, is beneficial for the motor performance of a child. However, the results of the studies included in this review were often difficult to compare, as relatively few studies were performed, and only a few studies investigated similar variables. Furthermore, children older than 6 years and social factors within the school and sports setting were only scarcely researched, which makes it hard to draw firm conclusions in these areas. Clearly, more research is warranted to further unravel the relationship between the social environment and motor skills, with a specific focus on 6–12-year-old children and environments outside of the home environment.

## Figures and Tables

**Figure 1 ijerph-18-07516-f001:**
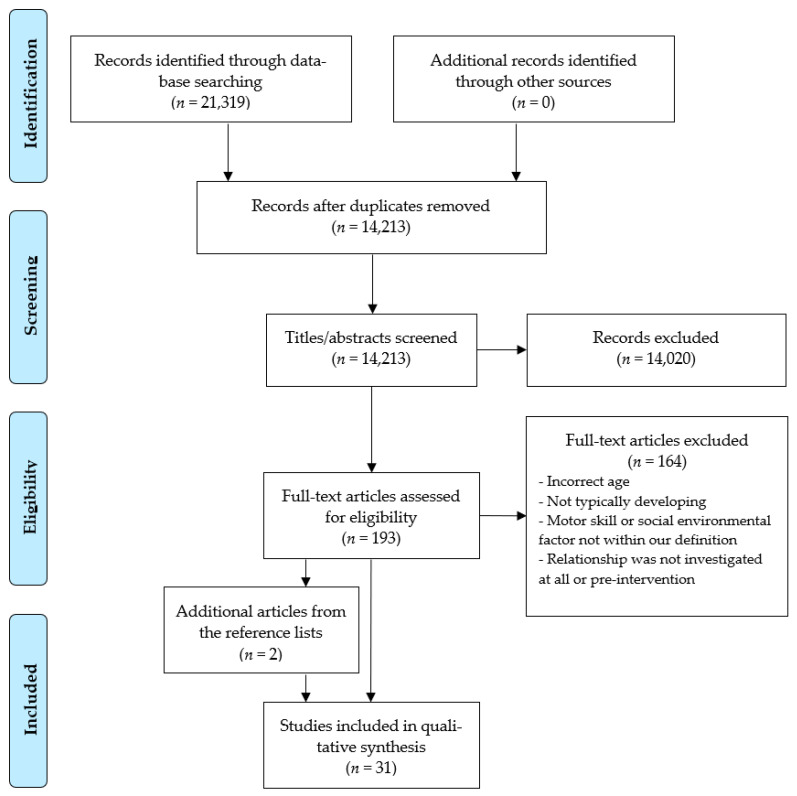
PRISMA flow diagram [25] of the study selection process to identify eligible studies for the systematic review.

**Figure 2 ijerph-18-07516-f002:**
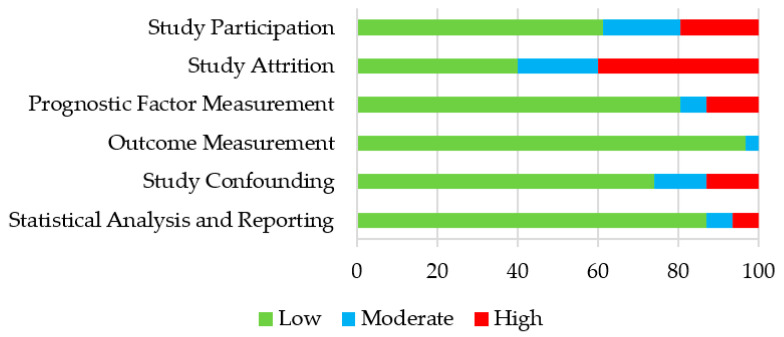
Percentage of included studies with a low, moderate, and high risk of bias assessed with the Quality of Prognosis Studies in Systematic Reviews (QUIPS) tool [27,28]. All the domains were assessed for all 31 studies, except the Study Attrition domain, which was assessed for five included studies.

**Figure 3 ijerph-18-07516-f003:**
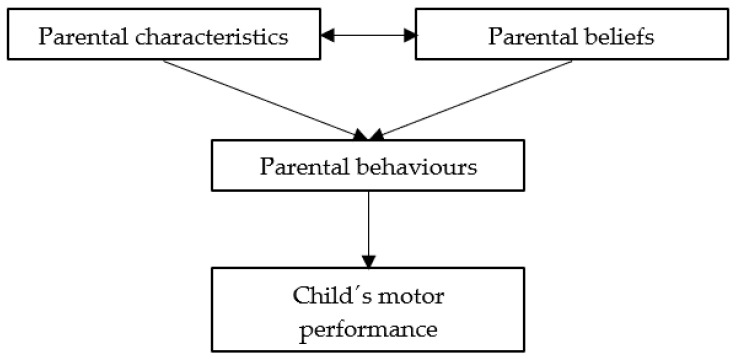
Framework of the relationship between the home learning environment and motor performance [62].

## Data Availability

The data is contained within the article.

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
