# Peer review of "The Relationship between Social Environmental Factors and Motor Performance in 3- to 12-Year-Old Typically Developing Children: A Systematic Review"

_ijerph, 2021, doi:10.3390/ijerph18147516_

Round 1
Reviewer 1 Report
Overview
This article presents a systematic review on an interesting and scarcely studied topic, which is the relationship between the social environment and the motor performance of developing children. They showed that parental beliefs regarding play, PA and motor development are associated with better motor performance, as well as some parental behaviors in child-rearing and PA practice in a sex related way. Other social environment factors although less investigated seem to have a relation such as spending time with older children, being appreciated by peers, or having interactions with teachers. The authors point out different gaps in the literature, and especially the lack of studies concerning 6-12 year-old children. However, when considering the search equation they have used in the databases, they have search motor performance using exclusively the term “motor skill” and it’s derivate in PUBMED and PSYCHINFO, “motor performance” and it’s derivate in EMBASE and “psychomotor skill” and it’s derivate in ERIC. This has led to ignoring a part of the literature on the related subject. In fact, the term “physical literacy” could have been added in the search equation. As a incomplete example, here are some reference that should be added to this review and would partly fill the gap in the above mentioned age-group:
- Law B, Bruner B, Scharoun Benson SM, Anderson K, Gregg M, Hall N, Lane K, MacDonald DJ, Saunders TJ, Sheehan D, Stone MR, Woodruff SJ, Belanger K, Barnes JD, Longmuir PE, Tremblay MS. Associations between teacher training and measures of physical literacy among Canadian 8- to 12-year-old students. BMC Public Health. 2018 Oct 2;18(Suppl 2):1039. doi: 10.1186/s12889-018-5894-7. PMID: 30285690; PMCID: PMC6167764.
- Platvoet S, Pion J, de Niet M, Lenoir M, Elferink-Gemser M, Visscher C. Teachers' perceptions of children's sport learning capacity predicts their fundamental movement skill proficiency. Hum Mov Sci. 2020 Apr;70:102598. doi: 10.1016/j.humov.2020.102598. Epub 2020 Mar 5. PMID: 32217216.
- Ha AS, Chan W, Ng JYY. Relation between Perceived Barrier Profiles, Physical Literacy, Motivation and Physical Activity Behaviors among Parents with a Young Child. Int J Environ Res Public Health. 2020 Jun 21;17(12):4459. doi: 10.3390/ijerph17124459. PMID: 32575873; PMCID: PMC7345247.
Minor Comments
Introduction
P2; line 71 : “… but the social environment continues to develop as well” should be reworded to be more explicit for something like “… but the social environment evolves as well”
P2; line 75: “trainers” or “coachs” should be added to “teachers and peers” since these terms have been used in the search equations used in the databases and present a part of the potentially influencing social environment
Results
P14; line228-244: this section should be slightly reorganize: in the introduction of this section, the author present 3 main parental characteristics related to family structure (“status of the parents”, “composition of the family” and “involvement of the father in the child-rearing”). However, they don’t give any detail on the last factor. Instead of that, they detail another factor which is “parental and maternal status” on the same section. Thus, “involvement of the father in the child-rearing” should be either withdrawn or further detailed, and “parental and maternal status” should be presented in another section since it is not part of “family structure”.
P15; line 294-307: this section should be slightly reorganized to follow the same logic as the preceding sections, starting with presenting the most represented factors. From line 296 to line 299, at the beginning of the section is presented a single study on the topic of “maltreatment and abuse of the child”. This study should be presented at the end of the “parental behaviors” section since it is isolated and presenting it at the front of the section puts it inappropriately in the light (it is not discussed later on with reason).
Discussion
It would be interesting to mention that no study has been found on the association between trainer or coach behavior or belief and children’s motor performance, probably due to the lack of studies in 6-12 year old children that have been reviewed, and that it is a gap in the literature.
P18; line 439-444: The sentence has to be cut in several sentences for more clarity.
P20; line 553-555: the assertion that “a higher amount of maternal PA to be related to better gross motor performance of 7- to 9-year-old girls » should be tempered since it was not the case in the Cools et al (2011) but only on Luz et al. (2018), and the reference should be mentioned appropriately.
P20; line 561: “oh” missing for “the presence of siblings”
P22; line 671-674: the term “forgotten years of development” should be lessened or at least discussed since this age group have been more studied regarding to children’s motor development with the increased number of research around the concept of physical literacy in the last 8-10 years.
Reviewer 2 Report
I appreciate the opportunity of reviewing this paper addressing such an interesting and relevant topic, the relationship between environment and motor performance. I think the information contained in this paper is potentially useful for clinicians, parents and education professionals.
I have some suggestions that may help to improve the quality of the paper.
Abstract: I find the information in lines 24-28 a bit confusing. 31 studies is by far not scarce. And the affirmation in line 27 is not really “informative”, a stimulating enviromente in terms of interaction with others is really vague
Introduction:
Line 34: “during childhood” is repeated twice in the pharagraph.
Please, find a synonym for “motor performance” as is overused through the text.
Line 81: As is a systematic review and not a experimental study, please add to the sentence, identify “in scientific literature”.
Line 98-109: Please rearrange this section in a more schematic disposition. Make a clear difference between inclusion and exclusion criteria, they appeared mixed in this section.
Line 101: Provide an example of which kind of definitions were considered as unfitting.
Results/Discussion:
Overall, I find that the authors have made a quite broad research question. 31 included studies is a huge number to analize, and later in the description of the different domains assessed, I have a strong feeling that maybe it would have been better to set more precise inclusion/exclusion criteria and narrow the analysis. For the reader, is too much information. I suggest that the authors reconsider their manuscript and limit their research interest either to the family environment or the school environment but not both.
Conclusions:
The conclusions are too vague, surely due to the large amount of studies analyzed, the broad range of age assessed…overall I find this review topic really interesting and necessary, so I encourage the authors to focus on some aspects of their interest and dismiss others or maybe even split the review into two manuscripts, as all the aspects they try to cover are equally interesting, but putting them altogether lowers the quality of the review.
Round 2
Reviewer 2 Report
After carefully reading the paper I think the authors addressed all my concerns, and made a great effort improving the manuscript. It is really a relevant and interesting topic and the limitations of the study had been addressed adequately. Their rationale is much clear now despite I still feel that they have been too ambitious trying to encompass the whole environment of young children. Nevertheless, the topic deserves visibility from my point of view.